



# Carbon dioxide dynamics in an agricultural headwater stream driven by hydrology and primary production

Marcus B. Wallin[1]*, Joachim Audet[2], Mike Peacock[3], Erik Sahlée[1], Mattias Winterdahl[1,4,5]

[1]Department of Earth Sciences, Uppsala University, Uppsala, Sweden

[2]Department of Bioscience, Aarhus University, Silkeborg, Denmark

[3]Department of Aquatic Sciences and Assessment, Swedish University of Agricultural Sciences, Uppsala, Sweden

[4]Department of Physical Geography, Stockholm University, Stockholm, Sweden

[5]Bolin Centre for Climate Research, Stockholm, Sweden

*Correspondence to*: Marcus B. Wallin (marcus.wallin@geo.uu.se)

**Abstract.** Headwater streams are known to be hotspots for carbon dioxide ($CO_2$) emissions to the atmosphere and are hence important components in landscape carbon balances. However, surprisingly little is known about stream $CO_2$ dynamics and emissions in agricultural settings, a land-use type that globally cover ca 40% of the continental area. Here we present

continuously measured in-situ $CO_2$ concentration data from a temperate agricultural headwater stream covering more than one year of open-water season. The stream $CO_2$ concentrations during the entire study period were generally high (median 3.44 mg C L$^{-1}$, corresponding to partial pressures ($pCO_2$) of 4778 µatm) but were also highly variable (IQR = 3.26 mg C L$^{-1}$). The $CO_2$ concentration dynamics covered a variety of different time-scales from seasonal to hourly, and with an interplay of hydrological and biological controls. The hydrological control was strong (although with both positive as well as negative

influences dependent on season) and $CO_2$ concentrations changed rapidly in response to rainfall and snowmelt events. However, during growing-season baseflow and receding flow conditions, aquatic primary production seemed to control the stream $CO_2$ dynamics resulting in elevated diel patterns. Given the observed high levels of $CO_2$ and its temporally variable nature, agricultural streams clearly need more attention in order to understand and incorporate these considerable dynamics in large scale extrapolations.

## 1. Introduction

Fluvial systems (streams and rivers) are estimated to dominate the inland water $CO_2$ source globally, surpassing $CO_2$ emissions by lakes and reservoirs by a factor of six (Raymond et al. 2013). However, this estimate relies on a number of assumptions and the scarcity of empirical data makes it uncertain. One of the critical gaps in the global upscaling is the lack of direct measurements from agriculture dominated areas (Osborne et al. 2010). Globally, agricultural land covers about 40% of the

total continental area (Ramankutty et al., 2008) but there are few studies specifically focusing on the magnitude and dynamics

of $CO_2$ emissions from agricultural streams. The few studies that do exist conclude that $CO_2$ concentrations in such streams

are generally high and up to five times greater than those in streams draining forested areas which are more extensively studied

(Borges et al. 2018; Bodmer et al. 2016; Wallin et al. 2018). For example, Bodmer et al. (2016) measured partial pressure of

$CO_2$ ($pCO_2$) in German and Polish streams and examined differences between forested and agricultural catchments. They

found that $pCO_2$ was generally 2-3 higher times in agricultural streams compared to streams draining forested areas. Similarly,

Borges et al. (2018) found high $CO_2$ concentrations in streams and rivers dominated by agriculture in the river system Meuse,

Belgium. They linked the higher $pCO_2$ in agricultural streams (up to 5 times higher than in forested areas) to elevated levels

of dissolved organic carbon (DOC), particulate organic carbon (POC) and inorganic nitrogen.

There are numerous factors influencing $CO_2$ patterns in stream systems and often site-specific controls dominate. Hence, large

scale generalizations are difficult to make (Crawford et al. 2017). $CO_2$ concentrations in nutrient-poor forest and peatland

streams are often related to variations in stream discharge but with site-specific response patterns, both positive and negative

(Crawford et al. 2017; Dinsmore et al. 2013). These response patterns have often been connected to the catchment

characteristics and changes in hydrological pathways, which in turn control the dominant $CO_2$ source areas of catchment soils

(Campeau et al. 2018; Leith et al. 2015; Dinsmore and Billett 2008). In contrast, other catchments lack a strong hydrological

control and instead display clear diel cycles in the stream $CO_2$ concentration indicating a metabolic control (Crawford et al.

2017). Here the interplay of photosynthesis and respiration (in-stream or terrestrial) could result in large day to night time

differences in stream $CO_2$. These recent findings concerning dynamics and controls on stream $CO_2$ concentrations have been

possible due to the development of cost-effective $CO_2$ sensors (e.g. Johnson et al. 2010; Bastviken et al. 2015) which have

enabled continuous data collection covering relevant time-scales. However, very little information about stream $CO_2$ dynamics

exists from agricultural areas, a land-use type that is heavily managed by humans from multiple aspects including hydrological

drainage, nutrient additions, soil cultivation etc. As a consequence, $CO_2$ patterns in agricultural streams could potentially be

very different than in other land-use types with amplified diel $CO_2$ dynamics due to high metabolism and/or quicker response

to hydrological events due to effective drainage systems.


In addition to the concentration gradient between the stream water and the above air, gas exchange is also highly dependent on the physical conditions at the air-water interface. For stream systems, the gas transfer velocity (often the variable given to describe the efficiency of the air-water gas exchange) is related to a combination of hydrological and morphological conditions of the stream channel, often including slope, velocity and depth (Raymond et al. 2012; Wallin et al. 2011). All these variables

are proxies for describing the water turbulence of the stream which controls the gas exchange but that is rarely directly measured (Kokic et al. 2018). Agricultural areas are often located in flat landscapes resulting in drainage systems that are low-gradient and slow-flowing (Rhoads et al. 2003; Hughes et al. 2010), conditions that prevent effective air-water gas exchange (Hall & Ulseth, 2019). However, whether the elevated $p$CO$_2$ observed in agricultural streams is an effect of land-use specific hydro-morphological stream conditions preventing efficient gas exchange or an effect of high internal (aquatic) or external

(terrestrial) CO$_2$ production is currently unknown.

Although recent studies have shown the potential importance of agricultural streams, there are still large knowledge gaps to be filled in order to improve our understanding concerning the influence of these waterbodies in landscape C cycling. Here we present high-resolution CO$_2$ concentration measurements in a Swedish agricultural headwater stream during more than a year

of open water season. The study aimed to 1) quantify CO$_2$ concentration levels in an agricultural stream and explore its temporal dynamics, 2) identify the main drivers causing temporal variability in stream CO$_2$ concentration and how they might vary with season.

## 2. Methods

### 2.1. Study area

The study was conducted within the 11.3 km$^2$ Sundbromark (SBM) catchment (59°55′N, 17°32′E), located 5 km NW of the city of Uppsala, Sweden (Figure 1). The 30 year (1960-1991) mean annual, January and July temperatures for the area are 5.3°C, -4.5 and 16.0 and with a mean annual precipitation of 535 mm (SMHI). The catchment is dominated by agricultural land (86%) and with minor influence of forest (8%) and urban areas (6%). The area is flat with only 28 m elevation difference from 41 m.a.s.l. at the highest point to 13 m.a.s.l. at the catchment outlet (Table 1). The bedrock consists of gneissic granites

and the soils are dominated by post glacial clay at lower elevations and with some influence of glacial clay and silt at higher

elevations. Although the bedrock does not contain any known carbonates, the soils are alkaline due to glacial carbonate

containing deposits resulting in a stream pH ranging 7.4-8.4 (Table 2) and with high electrical conductivity (EC, ranging 791-

1908 $\mu$S cm$^{-1}$) (Osterman 2018). The nutrient and DOC levels of the stream water (Table 1) are at the lower end of monitored

agricultural catchments in Sweden (Linefur et al. 2018; Kyllmar et al. 2014). The oxygen conditions are mainly undersaturated

(median D.O. = 53%) during the growing season. The arable fields are to a large extent artificially drained with extensive tile

drainage pipe systems connected to the stream network. The catchment is a part of the hydro-meteorological observatory

Marsta that was established in the late 1940s (Halldin et al. 1999).

To explore how representative the SBM catchment is for streams draining agricultural areas in the region, a snapshot sampling

survey was performed across 10 streams (denoted region UPP 2 in the study by Audet et al. (2019)) of various sizes (catchment

area 8.5-740 km$^2$) and agricultural influence (30-86%) distributed within a radius of 10 km from the city center of Uppsala

(Table S1).

### 2.2. Field sampling and analysis

The measurements were conducted during the open-water season from Sep 26, 2017 to Dec12, 2018. Stream $CO_2$ concentration

was monitored using an EosGP sensor (Eosense, Dartmouth, Canada). The sensor was covered by copper tape in order to avoid

biofouling. Sensor accuracy is <1% of the calibrated range (0-2% $CO_2$) + 1% of the reading corresponding to a maximum error

of ca 0.3 mg C L$^{-1}$ based on the maximum $CO_2$ measured in the current study. The $CO_2$ sensor was calibrated against known

gas standards before and after deployment. No significant drift (exceeding the above given uncertainty) in the instrument was

observed during the period. Volume fraction outputs from the sensor were corrected for variations in temperature and pressure

(atmospheric and water depth) using the method described in Johnson et al. (2010) and expressed in the unit of mg C L$^{-1}$. Only

$CO_2$ data measured at discharge rates > 0 L s$^{-1}$ were used in the analysis of the data.

Water level, water temperature and EC were measured together with $CO_2$ concentration at a V-notch weir. Water level was

measured using a pressure transducer (1400, MJK Automation, Sweden) mounted in a stilling well representing the stream
water level at the V-notch weir. Discharge was calculated from a stage-discharge rating curve based on a series of manual

measurements and according to a rating curve presented in Holmqvist (1998). Water temperature and EC were monitored

using a thermocouple (Type T) and a CS547A-L conductivity sensor (Campbell, UK), respectively. The sensors (except for

the pressure transducer) were deployed under the water surface attached to a wooden rod in the center of the stream just

upstream of the weir. All sensors were connected to a CR1000X data logger (Campbell, UK) which stored average data

(measurements every 1 minute) at a temporal resolution of 30 (in 2017) or 60 (in 2018) minutes.


Stable isotopic analysis of the dissolved inorganic carbon (DIC) ($\delta^{13}$C-DIC) was performed on six occasions during the falling

limb of the snowmelt discharge peak in 2018 in order to explore the temporal variability in DIC source. Samples for analysis

of $\delta^{13}$C-DIC were taken in a 60 ml glass vial completely filled with stream water and closed airtight with a rubber septum

below the water surface. In order to preserve the sample, 1 ml of highly concentrated $ZnCl_2$ solution was injected in each

sample (with subsequent release of 1 ml of sample in order to keep atmospheric pressure) directly after sample collection.

Samples were kept cold and dark until analysis. Prior to analysis, 2 ml of sample was injected into 12 ml septum-sealed pre-

combusted glass vials (Labco Limited) pre-filled with $N_2$ gas, and pre-injected with 1 ml of phosphoric acid in order to convert

all DIC species to $CO_2$(g) (Campeau et al. 2017a). The samples were analyzed using an isotope ratio mass spectrometer (DeltaV

Plus,Thermo Fisher Scientific, Bremen, Germany) Gasbench II (Thermo Fisher Scientific, Bremen, Germany) measuring the

$CO_2$ in the headspace. Each sample was analyzed seven times and the first two injections for each sample were discarded to

avoid memory effects, and the mean was taken of the other five to give the final result. The $\delta^{13}$C-DIC values are given in terms

of deviation from the standard Pee-Dee Belemnite (PDB) in per mille where R is the isotopic ratio of $[^{13}C]/[^{12}C]$:

$$\delta^{13}C\text{–DIC (‰)} = (R_{sample}/R_{standard}\text{-}1) \times 1000$$

Precipitation, air temperature and incoming shortwave (global) radiation data (Figure 2) were obtained from the Marsta

meteorological observatory located within the catchment ca 2.5 km from the stream sampling station (Halldin et al. 1999). In

the absence of direct measurements of photosynthetically active radiation (PAR) shortwave incoming radiation was used as a

proxy for available photosynthetic light.

A spatial sampling campaign for $CO_2$ concentration, pH, EC and water temperature was conducted on June 21, 2018 across

ten streams located around the city of Uppsala. The sampling was performed between 10.00 and 14.00 during the day. Samples

for $CO_2$ analysis were collected using the headspace method (Hope et al., 2004; Kokic et al. 2015). Briefly, 30 mL bubble-free

water were collected in 60 mL polypropylene syringes and equilibrated with a known volume of ambient air by shaking

vigorously for 1 min. The equilibrated headspace (15-20 mL) was recovered and analysed on an Ultraportable Greenhouse

Gas Analyzer (UGGA) (Los Gatos Research, USA) equipped with a soda lime filter and manual injection port. In situ $CO_2$

concentration was calculated from the UGGA-determined ppm values using Henry's law considering stream temperature

(Weiss 1974), atmospheric pressure, the added ambient air, as well as the water-air volume ratio in the syringe. pH, EC and

water temperature were measured in-situ in the streams with handheld instruments, for pH with a pH110 pH-meter (VWR,

USA), and for EC and temperature with a HI 99300 (Hanna Instr., USA).

### 2.3. Delineation of the stream network and catchment characteristics

Catchment area and characteristics were calculated in QGIS 3.8 based on a high resolution ($2 \times 2$ m) digital elevation model

(DEM) derived from LIDAR data (GSD Elevation data, grid 2+, Swedish Land Survey). Land use distribution within the

catchment was derived from the CORINE Land Cover 2018 product (European Environment Agency), and soil and bedrock

characteristics were based on digital versions of the Quaternary deposits (1:25,000 – 1:100,000) and bedrock (1:50,000 –

1:250,000) maps (Swedish Geological Survey).

### 2.4. Data analysis

The continuous data from the SBM catchment was divided into four periods (Autumn, Snowmelt, Spring and Dry period)

according to distinct phases in the hydrograph in order to further analyze the control on stream $CO_2$ concentration (Figure 3).

The stream $CO_2$ dynamics observed among the different periods were examined visually and any hydrological controls on the

$CO_2$ were identified by the presence and direction of $CO_2$-discharge hysteresis loops (Evans and Davies, 1998). Similar

hysteresis analysis was used to investigate diel patterns in the $CO_2$ concentration data. Spearman's rank correlation coefficient

was used to test for monotonic relationships between the diel amplitude in stream $CO_2$ concentration and potential drivers.

Correlations were considered significant if p < 0.05. The software JMP 14.2.0 (SAS Institute Inc., Cary, NC, USA) was used for all statistical calculations.

## 3. Results

The mean air temperature and total precipitation for the entire period (Sep 26, 2017-Dec 12, 2018) were 6.8 °C and 704 mm, respectively. The summer and autumn of 2018 were dry with generally low precipitation, the exception was on July 29 with 82 mm rain within 24 hours (Figure 2). Mean and median stream discharge for the open-water period were 30.6 and 0.9 L s$^{-1}$, respectively, and with a total range from 0 to 668 L s$^{-1}$ (corresponding to a range from 0 to 5.0 mm day$^{-1}$). However, due to high water table exceeding the range of the pressure transducer the absolute peak discharge occurring during April 5 to April

7 was missed in the measurements. The large skewness between mean and median discharge was an effect of the large number of days without waterflow over the weir during the summer and autumn 2018, 128 days (38%) out of the open-water period. According to frequency analysis, 67% of the days had a mean daily discharge <5 L s$^{-1}$. Despite the few days with discharge >100 L s$^{-1}$ (7% of the entire period), those days accounted for 69% of the accumulated discharge. The majority (84%) of these high discharge days occurred during the snowmelt in April.

### 3.1. General CO$_2$ patterns


The stream CO$_2$ concentrations during the entire study period (median and mean 3.44 mg C L$^{-1}$ and 3.94 mg C L$^{-1}$, respectively, corresponding to a $p$CO$_2$ of 4778 µatm and 5324 µatm) were highly variable (IQR = 3.26 mg C L$^{-1}$) (Figure 3) and displayed a bimodal distribution with frequency peaks at ~2.7 mg C L$^{-1}$ and ~6.1 mg C L$^{-1}$ (Figure S3). The lower peak was associated with the snowmelt and spring period, whereas the higher peak was attributed to the autumn period 2017 and to rain events

during the dry period of summer/autumn 2018. In addition to the bimodal shape a very distinct peak in frequently measured concentrations was observed at ~1.6 mg C L$^{-1}$. This peak was attributed to the minimum concentrations values for the diel cycles observed during the spring period.

### 3.2. Controls on stream CO$_2$ concentration

The autumn period started dry with low discharge (<3 L s$^{-1}$) for the initial month of measurements. The CO$_2$ concentrations

were at the same time highly dynamic but unrelated to variations in discharge. The CO$_2$ concentration reached the maximum

for the autumn (10.89 mg C L$^{-1}$, which was also the maximum for the entire study period) during late October followed by a decline in $CO_2$ to ca 2 mg C L$^{-1}$ in early November. During November and December four main rain events were identified which all displayed an increasing stream $CO_2$ concentration with increasing discharge. In three of these events a positive clockwise hysteresis loop was observed (Figure 4) where the $CO_2$ concentration reached its maximum before the discharge did. At the last event during the autumn 2017, the relationship between $CO_2$ concentration and discharge was close to linear, but still positive. During the snowmelt period the hydrograph was characterized by a diel cycle with melting during day-time resulting in daily discharge peaks which were suppressed during night-time freezing. In contrast to the autumn events the daily discharge peaks were negatively related to the stream $CO_2$ concentration, and with an anti-clockwise hysteresis loop where the minimum $CO_2$ concentration was reached before the highest discharge of the event (Figure 5). After the snowmelt discharge peak the spring and early summer periods (late April to early July) were dry with limited precipitation and with a steady decline in runoff (Figure 3). During this period the $CO_2$ concentration displayed a pronounced diel cycle with daily maximum and minimum $CO_2$ concentrations reached during early mornings (06:00) and late afternoons (18:00), respectively (Figure 6). The medium amplitude of the diel $CO_2$ cycle for this period was 2.03 mg C L$^{-1}$, corresponding to $pCO_2$ = 2974 µatm (IQR = 1.23 mg C L$^{-1}$, corresponding to $pCO_2$ = 2212 µatm), and with the size of the diel $CO_2$ concentration amplitude being related to both the daily mean water temperature and the shortwave radiation (Figure 7). The diel pattern displayed a clear negative anti-clockwise $CO_2$-streamwater temperature hysteresis loop, where the median $CO_2$ concentration could differ up to 75% between day and night-time although being measured at the same stream water temperature (Figure 8).

From early July the stream dried out and hence no runoff over the V-notch weir was generated. During this period the $CO_2$ sensor was mostly recording an atmospheric signal. However, for five rain events during the summer and early autumn runoff was generated which allowed stream $CO_2$ determination for shorter periods (Figure 9). During these runoff events (< 2 days long) high $CO_2$ concentration pulses were recorded (up to 11 mg C L$^{-1}$). At all events $CO_2$ was recorded for a longer period than the discharge as the small dam above the v-notch weir was still water-filled for some time after runoff over the weir ceased. Also, common for all events was that the stream $CO_2$ concentration continued to increase although the discharge peak had passed. During July 29 a heavy rain storm occurred with 82 mm precipitation during 24 hours. Although more than 15%

of the long-term annual mean precipitation fell during one day, low discharge was generated (maximum discharge 6.1 L s$^{-1}$) due to high evapotranspiration and dry soils (Figures 3 and 9). However, the rainstorm event resulted in close to the highest stream $CO_2$ concentration (10.81 mg C L$^{-1}$) being observed during the studied period. As soon as the stream was more permanently refilled in early December and with discharge generated over the weir, the stream $CO_2$ concentration was back to

similarly high levels (typically 5-8 mg C L$^{-1}$) as observed in the autumn of 2017.

### 3.3. Sources of DIC

The $\delta^{13}$C-DIC data collected during the falling limb of the spring discharge peak (discharge range 130-9.6 L s$^{-1}$) were ranging from -13.8 to -12.2‰. This narrow range suggests a relatively constant source of inorganic C during the spring period. Although there was a tendency towards more negative $\delta^{13}$C-DIC values at higher discharge, no significant relationship was

found (Figure 10). $\delta^{13}$C-DIC was also unrelated to the stream $CO_2$ concentration (data not shown).

### 3.4. Spatial representativeness

The ten streams manually sampled around Uppsala displayed a wide range in $CO_2$ concentrations (1.8-4.6 mg C L$^{-1}$) on the day of sampling (June-21 2018), and with the SBM stream (site 3 in table S1) being close to the overall median (SBM, 2.7 mg C L$^{-1}$; overall median, 3.0 mg C L$^{-1}$) (Table S1). Furthermore, the $CO_2$ concentration manually sampled at SBM was close to

the sensor recorded $CO_2$ (2.59 mg C L$^{-1}$) at the hour of sampling. The SBM stream was also close to the spatial median DOC concentration but slightly elevated in $NO_3$ and $PO_4$. The $CO_2$ concentration was on a spatial scale related to pH but unrelated to catchment area or land-use distribution within the catchment. Furthermore, the $CO_2$ concentration was on a spatial scale unrelated to open-water mean values of DOC, $PO_4$ and $NO_3$, although these variables were sampled during a different period than the $CO_2$.

### 4. Discussion

In order to produce large scale estimates of the exchange of GHGs between inland surface waters and the atmosphere, a basic requirement is to know the aqueous concentrations of the gases of interest and how they might vary over time. Headwater streams have been identified as "hotspots" for $CO_2$ emissions (Raymond et al. 2013; Wallin et al. 2018), but there is limited data capturing the temporal resolution, specifically from streams draining agricultural regions, making large scale

generalizations uncertain. Due to effective drainage, high nutrient conditions and high sun-light exposure (due to zero/limited

        tree cover), agricultural streams could potentially be very different in their $CO_2$ dynamics compared with streams draining

        other environments. Here we continuously measured stream $CO_2$ concentration in a headwater catchment dominated by

        agricultural land-use (86%) covering more than one year of the snow-free period. In line with findings from similar studies

        from other environments (arctic tundra, boreal forest, temperate peatlands, alpine) (e.g. Rocher-Ros et al. 2019; Riml et al.

2019; Crawford et al. 2017; Peter et al. 2014; Dinsmore et al. 2013) we found a mixture of controls on stream $CO_2$ operating

        at different time-scales generating a highly dynamic stream $CO_2$ pattern. These time-scales covers seasonal patterns to diel

        cycles, or even shorter scales associated to discharge events. Both the magnitude of $CO_2$ concentrations, and their associated

        temporal dynamics were found to be high in the current agricultural stream when compared with the literature. The mean $CO_2$

        concentration (3.94 mg C $L^{-1}$ corresponding to a $pCO_2$ of 5324 µatm) is at the high end when compared with other high-

frequency $CO_2$ data sets covering low-order (<3rd stream order) catchments draining multiple environments, including arctic

        tundra, boreal forest, hemi-boreal forest, temperate forest, temperate peatlands and alpine areas (typically ranging from ca 0.2

        to 6 mg C $L^{-1}$) (Crawford et al. 2017; Natchimuthu et al. 2017; Peter et al. 2014; Dinsmore et al. 2013). Still, $CO_2$ concentrations

        in SBM do not seem to be exceptionally high compared to snapshot-based data from other agricultural streams.

The spatial variability seen in this study, although only based on snapshot samples, and previous studies indicate that $CO_2$

        concentrations in agricultural streams are comparably high (Borges et al. 2018; Bodmer et al., 2016; Sand-Jensen & Staehr,

        2012). In addition, the observed temporal dynamics presented here are, to our knowledge, among the most pronounced in the

        literature, although the number of high-frequency stream $CO_2$ data sets are limited. For example, the rapid decrease in stream

        $CO_2$ during the autumn of 2017, the strong diel cycle (diel amplitude <5.0 mg C $L^{-1}$) during the spring/early summer period,

or the rapid and high $CO_2$ pulses (<11.0 mg C $L^{-1}$) occurring in accordance to rain events during the dry late summer/autumn

        period. These high $CO_2$ dynamics clearly illustrate the need for continuous high frequency $CO_2$ concentration measurements

        in streams in general, and in agricultural streams more specifically. Without such high-frequency data, representative estimates

        of agricultural stream $CO_2$ will be associated with high uncertainty. Although based on measurements from a single stream,

**Biogeosciences**
**Discussions**

these findings in turn indicate that current large-scale stream $CO_2$ emission estimates, which are largely based on snapshot

concentration data with low (or no) resolution in time, might be specifically uncertain for agricultural regions.

According to our continuous data the highly dynamic pattern in stream $CO_2$ concentration is driven by a complex interplay of

hydrology and biology. The high autumn concentrations observed both in 2017 and 2018 are likely an effect of high respiration

of organic matter in the stream channel and/or in the adjacent soil water (Figure 3c). This is supported by efficient aquatic

microbial DOC degradation ($<800$ µg C L$^{-1}$ d$^{-1}$) observed during the autumn period across the ten streams (agricultural land-

use, 30-86%) included in the spatial sampling campaign (Peacock et al. unpublished 2019). This should be compared with

organic C degradation rates determined in boreal forest and mire streams displaying typically lower rates ($<300$ µg C L$^{-1}$ d$^{-1}$,

Berggren et al. 2009). The positive $CO_2$-discharge relationships indicated that event flow pathways, whether those are more

surficial or different spatially, were in contact with soils with higher concentrations of $CO_2$ compared to flow pathways during

base flow (Evans & Davies, 1998; Seibert et al., 2009). Also, the clock-wise shape of the hysteresis loop suggests that there

is a buildup of $CO_2$ in the catchment that is flushed out during rain events (Figure 4). The $CO_2$ pool seems to be limited as the

$CO_2$ concentration drops before the maximum discharge peak occurs, or that vertical patterns in the $CO_2$ soil profile control

the stream $CO_2$ dependent on dominating flow paths (Evans and Davies, 1998; Öquist et al. 2009). This could explain that the

stream $CO_2$ increase did not reach any source limitation at rain events of lower magnitude (Figure 4d). Similar positive $CO_2$

concentration-discharge patterns have been observed across different low-order streams (e.g. Crawford et al. 2017; Dinsmore

et al. 2013) but the absolute patterns are often concluded to be highly site-specific and even event-specific. Here we suggest,

by exploring the hysteresis loops, that such positive relationships are influenced by the size of the available catchment $CO_2$

pool or the hydrological connectivity to it. In a highly drained low-elevation agricultural landscape where much of the stream

runoff is generated through drainage pipes (Castellano et al. 2019), the extent and spatial distribution of these connections

between ground- and surface water are central for the $CO_2$ patterns observed in the stream.

In contrast to the patterns observed during the autumn, during the snow melt period the stream $CO_2$ was diluted at discharge

increases following a diel pattern (Figure 5). The melting and freezing between day and night-time suggests that melt-water





from the surface snowpack during day time to a larger extent reached the stream without picking up an elevated $CO_2$ signal.

Similar dilution patterns in conjunction with snowmelt have been observed in catchments of various land-use but specifically in peatland catchments with limited forest cover (e.g. Wallin et al. 2013). The similarity between this agricultural catchment and open peatlands could potentially be the effect of an efficient melting of the snowpack. Both non-forested peatlands and agricultural fields are open areas subject to direct sunlight, and wind and rain exposure, while the soil under the snow remains frozen. As a result, a large share of the melt-water will never infiltrate the soil but instead reach the surface drainage system

as overland flow (Laudon et al. 2007). This is further accompanied by the low hydraulic conductivity of clay soils, which are dominating the catchment of the current study. Although we did not capture the 2-3 days of peak spring flood (due to a water level out of the range of the pressure transducer) it was evident that the stream $CO_2$ concentration was diluted from ca 6.0 mg C L$^{-1}$ to ca 2.0 mg C L$^{-1}$ during these days, something that is further supported by the similar drop in EC during the peak spring flood from ca 900 to ca 150 µS cm$^{-1}$. However, as soon as the absolute discharge peak passed, the stream $CO_2$ concentration

recovered rapidly to the pre-peak levels suggesting a shift to hydrological pathways that mobilize a high $CO_2$ pool, again supported by the concurrent increase in EC. April and May 2018 were characterized by warm and clear weather with an average 4.2°C higher air temperature and 255 more sun hours than the 30-year mean (1961-1990, SMHI). Altogether, this stimulates a kick-start of the aquatic primary production upon snowmelt, which likely explains the steady decline in $CO_2$ that occurred during late April/early May. During the spring and early summer, a strong diel pattern in $CO_2$ concentration further

developed, likely driven by aquatic primary production consuming $CO_2$ during day-time. Such diel $CO_2$ patterns are commonly observed in stream $CO_2$ time series at base-flow or during receding flow conditions (e.g. Riml et al. 2019; Peter et al. 2014) and are especially pronounced in amplitude in nutrient-rich streams or in streams without canopy shading (Alberts et al. 2017; Crawford et al. 2017; Rocher-Ros et al. 2019). Initial evaluation of the δ$^{13}$C-DIC data collected during the spring period suggests a relatively steady mixture of geogenic and biogenic DIC although somehow related to variations in discharge (Figure

11). However, given the suppressed stream $CO_2$ during the spring period, together with the strong diel cycle caused by aquatic primary production, fractionation of a strict biogenic DIC pool (with a δ$^{13}$C-DIC from -28 to -20‰) could theoretically push the δ$^{13}$C-DIC towards the less negative values observed in the current study (from -13.8 to -12.2‰) (Campeau et al. 2017b).

Combined studies on aquatic metabolism, C dynamics and stable isotopic composition would further be recommended to disentangle the dynamic $CO_2$ source patterns in this type of agricultural system.


The spring, summer and autumn periods of 2018 were generally dry leading to the stream channel drying out during long periods. The rapid rewetting periods (< 2 days) that occurred following larger precipitation events resulted in high $CO_2$ pulses (3-11 mg C $L^{-1}$) generally exceeding the overall median level of stream $CO_2$ (3.44 mg C $L^{-1}$) observed during the study period. The intermittent nature of streams, with distinct drying and rewetting episodes, is known to generate high $CO_2$ concentration

pulses and subsequent emissions (Marcé et al. 2019). Such rapid pulses are generally suggested to be a result of intense respiration in the stream bed sediments upon rewetting, or due to a rapid mobilization of terrestrial C, both organic (DOC) and inorganic ($CO_2$) in connection to precipitation events. However, the findings of high $CO_2$ pulses upon rewetting have mostly been done in areas that display pronounced dry and wet seasons e.g. Mediterranean areas or Australia (e.g. Gomez-Gener et al. 2015; Looman et al. 2017). Here we show that such stream intermittency can also cause high and rapid $CO_2$ pulses in a

Swedish agricultural setting, highlighting the need for expanding the geographical coverage of studies that investigate stream intermittency in relation to GHG dynamics and emissions. An obvious tool in this work is the use of continuous sensor-based measurements which allow capturing the episodic and unpredictable nature of these phenomena.

## 5. Conclusions

It is evident from the current study that the stream $CO_2$ dynamics in an agricultural headwater catchment are highly variable

across a variety of different time-scales and with an interplay of hydrological and biological controls. The hydrological control was strong (although with both positive as well as negative influences dependent on season) and rapid in response to rainfall and snowmelt events. However, during growing-season baseflow and receding flow conditions, the aquatic primary production seems to control the stream $CO_2$ dynamics, which in turn sets the basis for atmospheric emissions. Given the observed high levels of $CO_2$ and its temporally variable nature, agricultural streams clearly need more attention in order to understand and

incorporate these considerable dynamics in large scale extrapolations.



## 6. Data availability

Data is available from the Uppsala University data repository, LINK WILL BE ADDED

## 7. Author contribution

MBW and MW brought the idea and designed the study. MBW funded and instrumented the catchment and analysed the data.
MW conducted the GIS analysis. JA, MP and ES provided ideas and data. MBW wrote the manuscript with great support from all co-authors.

## 8. Competing interests

The authors declare that they have no conflict of interest

## 9. Acknowledgements

Financial support to MBW from the King Carl-Gustaf XVI award for environmental science and from the Finn Malmgren foundation is acknowledged. JA was supported by FORMAS (grant 2015-1559). Jacob Smeds, My Osterman, Philip Johansson and Maud Oger are acknowledged for great support in field and lab.

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



**Table 1. Catchment characteristics of the SBM catchment**

| | |
|---|---|
| Catchment area (km$^2$) | 11.3 |
| Elevation range (masl) | 13-41 |
| *Landuse distribution (%)* | |
| Agricultural land | 86 |
| Forest | 8 |
| Urban | 6 |
| *Main Soil type distribution (%)* | |
| Post glacial clay | 48 |
| Glacial silt | 22 |
| Glacial clay | 14 |
| Sandy till | 12 |
| *Main bedrock distribution (%)* | |
| Granodorite granite | 89 |
| Tonalite granodiorite | 6 |
| Dacite rhyolite | 3 |
| Granite | 2 |





**Table 2. Water chemistry at the outlet of the SBM catchment collected during June-November 2017 (n = 8) (Osterman 2018).**

|  | Median | Mean | Min-Max |
|---|---|---|---|
| pH | 7.7 | 7.8 | 7.4-8.4 |
| EC ($\mu S\ cm^{-1}$) | 1082 | 1273 | 791-1908 |
| $NH_4$-N ($mg\ L^{-1}$) | 0.10 | 0.08 | 0.01-0.1 |
| $NO_3$-N ($mg\ L^{-1}$) | 0.7 | 1.9 | 0.09-6.5 |
| $PO_4$-P ($mg\ L^{-1}$) | 0.07 | 0.09 | 0.01-0.2 |
| DOC ($mg\ L^{-1}$) | 10.0 | 9.6 | 4.2-13.1 |
| D.O. (%) | 53 | 62 | 31-119 |




**Figure 1. Land use distribution within the SBM catchment (GSD elevation data, grid 2+, ©Swedish Land Survey; CORINE Land Cover 2018, European Environment). The stream-based measurements were conducted at the catchment outlet (red dot) whereas the meteorological data derived from the Marsta Observatory (black dot).**



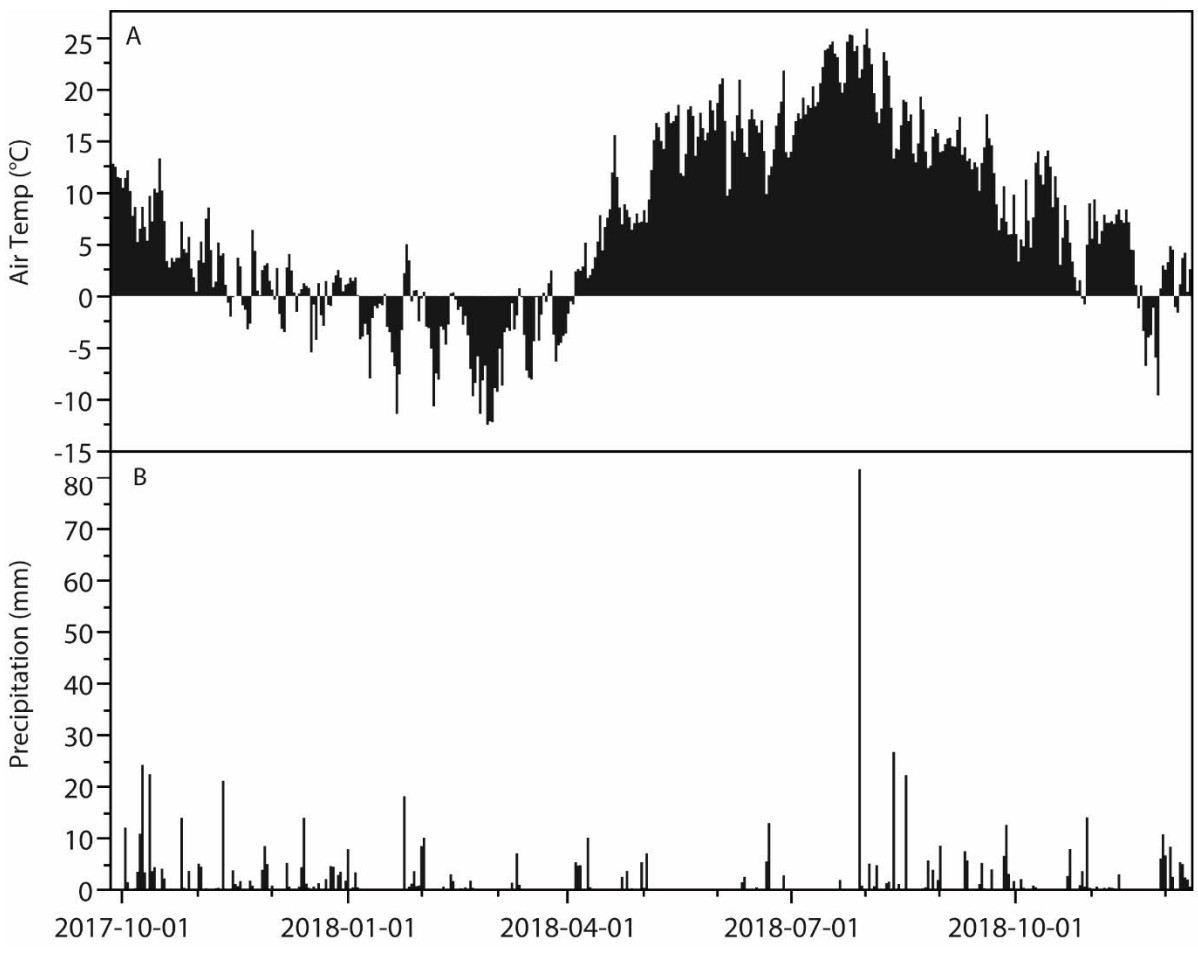

**Figure 2. (A) Daily mean air temperature, and (B) daily precipitation during the study period (Sep 26, 2017-Dec 12, 2018) at the Marsta Observatory. Due to malfunctioning sensor the precipitation data for July 29 2018 is collected from the nearby (3 km) SMHI station, Ärna.**






**Figure 3.** Time series of A) stream discharge (Q) with sampling days for δ¹³C-DIC highlighted by red dots, B) stream water temperature, C) electrical conductivity (EC), and D) CO₂ concentration for the study period Sep 26, 2017-Dec 12, 2018, with break for the ice- and snow-covered period December-March. The CO₂ data include periods when the sensor was above the water surface during dry periods in summer/autumn of 2018.




**Figure 4. Stream CO₂ concentration (black) and discharge (red) for the autumn 2017 period with CO₂-Q hysteresis plots for four rain events.**



**Figure 5. Stream CO₂ concentration (black) and discharge (red) for the snowmelt period 2018 with CO₂-Q hysteresis plots for four discharge events.**



**Figure 6.** Time series of (A) Stream $CO_2$ concentration (black) and discharge (red), and (B) water temperature (black) and shortwave incoming radiation (SR, red) covering the period April-July 2018. Note the reverse axis for shortwave 475 incoming radiation.





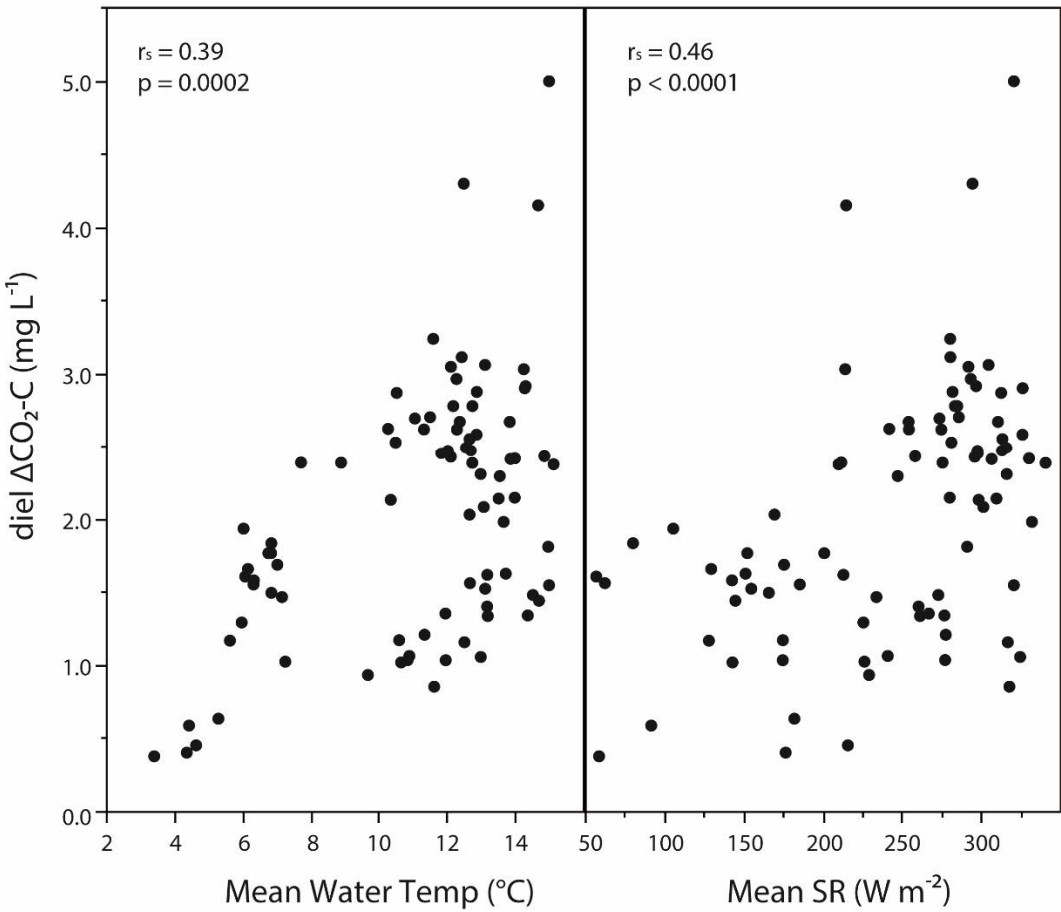

**Figure 7. Diel amplitude in stream CO₂ concentration in relation to A) daily mean stream water temperature, and B) daily mean shortwave radiation (SR), covering the period April-July 2018. Statistics are given according to Spearman's rank correlation.**





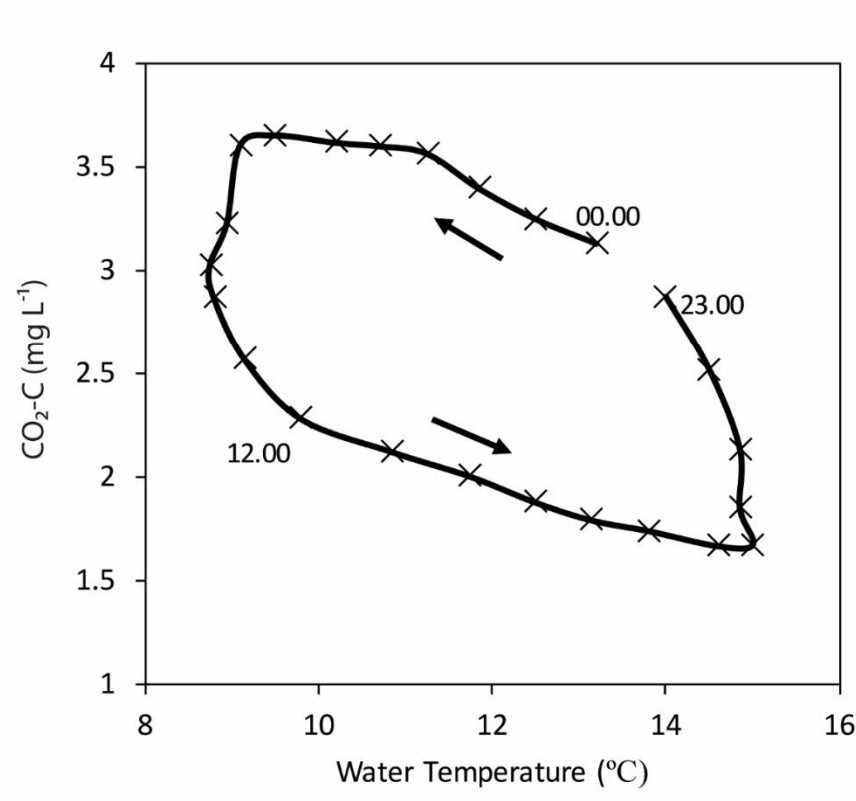

**Figure 8. CO₂-Water temperature hysteresis loop based on the median daily values presented in figure 7 covering the period April-July 2018.**





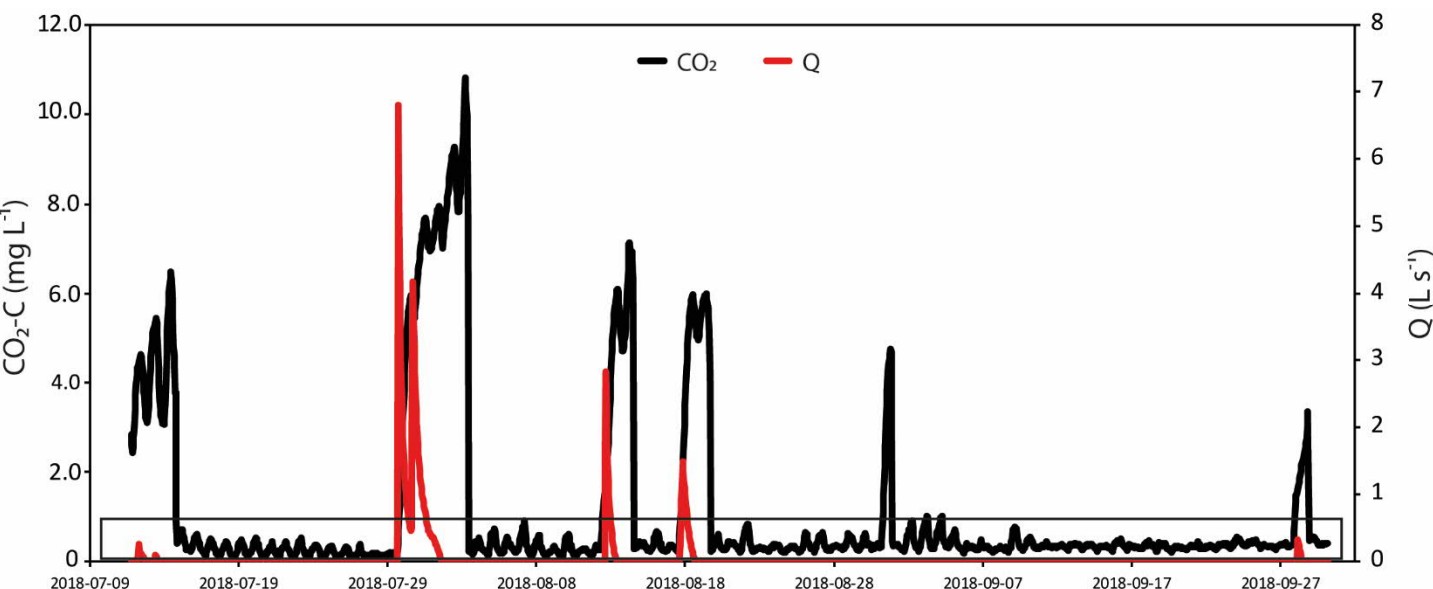


**Figure 9. Stream CO₂ concentration (black) and discharge (red) for the dry period (July-September 2018). Periods when the CO₂ sensor was above the water table capturing an atmospheric signal (i.e. with concentrations <0.5 mg C L⁻¹) are highlighted by the lower box.**






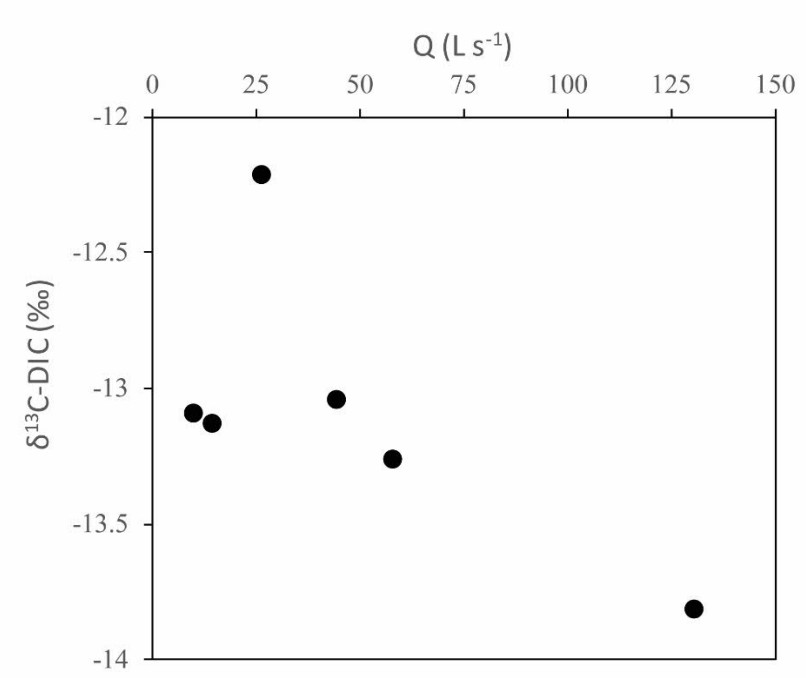

**Figure 10. δ¹³C-DIC as a function of stream discharge. The six sampling occasions covered the falling limb of the snowmelt peak April-June 2018.**