# Peer review of "Carbon dioxide dynamics in an agricultural headwater stream driven by hydrology and primary production"

_Biogeosciences, 2019_

## Referee Comment (RC1) · Anonymous Referee #1 · 3 Mar 2020

Headwater streams are known hotspots for $CO_2$ emissions, although studies of headwater streams draining agricultural catchments, and specially studies that includes a temporal dimension, are sparse. In this study, a headwater stream draining an agricultural catchment was continuously monitored during for approximately one year, and the responses in $CO_2$ concentrations to hydrological variations were studied.

General comments

This study provides important insights of $CO_2$ and discharge dynamics in a headwater stream draining a catchment impacted by agriculture. We need more studies like this in order to better understand the exchange of greenhouse gases between inland waters

and the atmosphere. Overall, I think the manuscript is very good. The study is well designed and presented in a well-structured way. I have only a few, although important, remarks that I think would improve the manuscript. Firstly, this paper would benefit from the authors emphasizing the relevance of their study better. For instance, this study points out potential effects of stream intermittency for streams draining agricultural catchments. This finding is highly important with respect to climate change. Despite this, the authors do no mention this neither in the abstract nor in the conclusions of the paper. Secondly, the manuscript would benefit from a more extensive discussion, for example how this stream compares to other agricultural influenced streams, if the type or insensitivity of the agriculture matters, land use change etc. Lastly, the readability of the manuscript could be greatly improved by simple sentence adjustments, such as shortening sentences and inserting more commas. Also, the figures could be designed in a more intuitive way.

Specific comments

Abstract

L15: It is unclear what "one year of open-water season" means. It would be helpful to add the dates and/or number of monitored days.

L22: I recommend the authors to add a sentence about the effects of indeterminacy of streams draining agricultural catchments here, since this is an important finding of the paper.

Introduction

L41-42: This sentence is unclear. What do you mean with positive and negative responses? Please clarify.

L44: "...dominant $CO_2$ source areas of catchment soils"? Please rephrase this sentence.

L45: Please specify what kind of other catchments.

L48: New paragraph needed.

L50: Please specify what "relevant" time-scales are.

L69: Please specify what you mean with high-resolution. Also, as mentioned before, it is unclear what "one year of open-water season" actually means.

Methods

L76: Please rephrase this sentence. Is it unclear if you mean the annual mean temperature or the January and July temperatures. This is especially important since you do not mention the precipitation in January or July - perhaps this could be added.

L82: Stream pH ranging between 7.4 and 8.4. Also, this sentence would be much more readable if you would add a comma. In general, I would recommend using commas more frequently.

L83-84: How much lower? Please provide a reference percentage.

L86-87: This sentence could be moved to the beginning of the paragraph.

L90: influences

L91: Table S1; Figure S2

L93: Would it be possible to add here the percentage that were snow/ice-free (and included in your study) as well as the percentage when the stream was falling dry?

L100: This is quite confusing for the reader, especially since you have not mentioned before that the stream is falling dry during some periods of the year. In general, I would recommend you to highlight the stream intermittency better, including adding some sentences in the introduction about this.

L108-109: Please clarify. What is the temporal resolution of your data?

L112-114: Please rephrase. Also, how many replicates?

L117: Please clarify. When was the phosphoric acid added?

L121: Did you run any standards?

L129: Add reference to Figure S2 here.

L140: It would probably be easier to follow if you move this paragraph to the beginning of the methods section.

L145: Another example of a sentence where the overall readability could be greatly improved if more commas are added.

Results

L155: This sentence is confusing. Precipitation is usually in mm/year however the period is for a bit more than a year. I assume that the "total precipitation" represent the precipitation for the whole period. Thus, it would be easier to read if the sentence would first state the mean air temperature (XXX) and then the total precipitation (XXX).

L205: It would be good to also add the corresponding $pCO_2$ here for reference.

L212: Same as above, the corresponding $pCO_2$ values would be helpful as reference values.

Discussion

L231: "highly dynamic pattern in streamwater $CO_2$ concentration".

L250: Please add references.

L256: could

L258-260: Please rephrase.

L266-270: Great paragraph. Would it be possible to develop more on this?

L271-272: Please rephrase.

L309-311: Another great paragraph. This could also be further developed and better highlighted.

Tables

Table 1: Throughout the manuscript, you write either "land-use" or "land use". In the table it is obviously a spelling mistake; however, please be consistent with the terminology throughout the whole manuscript.

Table 2: Would be good to add the name of the catchment and not only the abbreviation.

Figures

Figures: I recommend the authors to redo all figures. They are not intuitively designed or appealing for the reader.

Figure 7: Add regression line?

Figure 10: In the text it is written that d13C-DIC was NOT a function of Q?

---

## Referee Comment (RC2) · Anonymous Referee #2 · 11 Mar 2020

Inland waters and specifically headwaters emit significant amounts of $CO_2$ to the atmosphere; however, studies focusing in agricultural streams and including continuously measured in-situ $CO_2$ from are rather rare. In this MS, the authors continuously monitored $CO_2$ with cost-effective $Co_2$ sensors during one year and explored the spatio-temporal variations of $CO_2$ throughout the year as a function of hydrology and metabolism.

General comments The MS bg-2019-486 provides an interesting study about $CO_2$ dynamics in one stream draining a catchment largely dominated by agriculture. An important finding is that stream intermittency can cause rapid pulses of $CO_2$ even in

catchment with no pronounced dry and wet seasons. I think this is an important matter to better understand carbon emissions from streams at the global scale, in the context of climate change (change in hydrology). In line with this result, it could be useful to add somewhere in the discussion the spatial representativeness at the global scale of the stream studied here. In addition, it could be nice to add discussion/comparison of this agricultural stream with other agricultural streams worldwide, because the hydrology should be very different.To increase the readability, I suggest to better define some terms used in this study, particularly, open-water season, and the different periods, and also define better the time-intervals of these seasons throughout the text. Indeed, to my opinion, those terms are specific to boreal systems, and sometimes it is difficult to follow for a reader who is novice with boreal landscapes. A second important finding is the strong biologic control (aquatic primary production) of the CO2 dynamics during base flow that should decrease CO2 emissions during this period. Indeed, during base flow it is common to observed higher CO2 concentration in streams because deeper levels of groundwater are involved. Perhaps the authors could further developed this. Overall, I found the dataset very interesting; it is rare to have such continuous measurements for CO2 in streams. In addition, I found the paper well written. Perhaps the quality of some figures could be improved. Overall, I support publication of this manuscript and below are some more detailed comments.

Specific comments

Abstract L. 15-16: It would be nice for the reader adding the size of the catchment, the date of open-water season, and the time-step of CO2 measurements.

Introduction L.31-33: The authors can check this reference that suit with their study (Deirmendjian et al, 2019. Importance of the vegetation-groundwater-stream continuum to understand transformation of biogenic carbon in aquatic systems – a case study based on a pine maize comparison in a lowland sandy watershed), where the concentration of CO2 in agricultural and forested streams (and in groundwater) in a temperate catchment was compared. They found no differences between both streams

because degassing in agricultural streams was prevented. L. 43-45: Please clarify this sentence. You mean that different level of soils are exported in function of the change in hydrology? L-40-55: To my opinion, there is a lightly lack of spatial references in this paragraph. Indeed, I guess that agricultural streams in tropical or boreal areas are very different in terms of hydrology and carbon dynamics. Could you mention spatial references? L. 69: High-resolution: what is the time-step of measurements?

Methods L.78: What kind of cropland it is? This is important for the d13C-DIC L. 83: Lower end: how much lower? L. 85: Growing season: what is the time interval? L. 97: what was the concentration of gas standards? L.100: discharge rates lower than 0 L/s: so you mean when the stream was dry or when the stream was frozen? Or both? It is a bit confusing. L.101: Figure S1 L109: You wrote one measurements each minute but then a temporal resolution of 30. It is a bit confusing what is the meaning of temporal resolution here? L.120: What is the volume of the injections? L.129: Please specify that these streams were not located in your catchment and add the reference to the figure S2 L. 145: Please define better your four periods. What are the time intervals?

Results L.157: Please refer to figure 3 L.168-172: Please add corresponding pCO2 for reference, as you did L.166. To my opinion, I suggest to do that for the remainder of the text because pCO2 in ppmv is more "understandable" that CO2 in mg/L.

Discussion: L.225: I would not rush on conclusion about zero/limited tree cover along agricultural streams, at the global scale. I am agree considering your figure S2 that this is the case in your catchment. However, in temperate climate it is very common to observe riparian forest along agricultural streams.

Figures Figure 1: In the left part, I suggest to add a map of Europe rather than just Sweden. Please add a scale in the left part too. Figure 2: I suggest to separate the different periods (autumn, snowmelt, spring, dry period) with dotted lines, as you did in the next figure. Figure 4: It is not very intuitive what the time interval is for A, B, C and D. Figure 5: Same remark Figure 7: Perhaps add regression line with slope

---

## Author Comment (AC1) · 29 Mar 2020

**Reviewer (R#1) comments and author responses to ms bg-2019-486**

**Reviewer comments are given in normal style and with author responses in *italic***

Headwater streams are known hotspots for CO2 emissions, although studies of headwater streams draining agricultural catchments, and specially studies that includes a temporal dimension, are sparse. In this study, a headwater stream draining an agricultural catchment was continuously monitored during for approximately one year, and the responses in CO2 concentrations to hydrological variations were studied.

General comments

This study provides important insights of CO2 and discharge dynamics in a headwater stream draining a catchment impacted by agriculture. We need more studies like this in order to better understand the exchange of greenhouse gases between inland waters and the atmosphere. Overall, I think the manuscript is very good. The study is well designed and presented in a well-structured way. I have only a few, although important, remarks that I think would improve the manuscript.

*Response: We thank reviewer #1 for their overall positive evaluation of our manuscript and appreciate that it is found "very good" and "well designed and presented in a well-structured way". We believe that the revised manuscript has been significantly improved following the comments given by reviewer #1.*

Firstly, this paper would benefit from the authors emphasizing the relevance of their study better. For instance, this study points out potential effects of stream intermittency for streams draining agricultural catchments. This finding is highly important with respect to climate change. Despite this, the authors do no mention this neither in the abstract nor in the conclusions of the paper.

*Response: We agree that this is an important finding that we do not well enough lift up as one of the main messages. We do not have enough years of measurements for saying how common the intermittency of this specific stream is. The spring and summer of 2018 was unusually dry, but this kind of conditions are expected to occur more frequently in the future. We have in the revised version included this finding in the abstract and also more explicitly in the conclusions of the study.*

Secondly, the manuscript would benefit from a more extensive discussion, for example how this stream compares to other agricultural influenced streams, if the type or insensitivity of the agriculture matters, land use change etc.

*Response: We have in the revised version tried to improve the discussion on the spatial representativeness of our findings and also included references concerning DOC-discharge responses in agricultural areas.*

Lastly, the readability of the manuscript could be greatly improved by simple sentence adjustments, such as shortening sentences and inserting more commas. Also, the figures could be designed in a more intuitive way.

*Response: We have in the revised version tried to improve the readability of the text where appropriate, and we agree that some of the figures needed a quality lift up, although no suggestions on how were given by the reviewer. We have improved the quality of the figures*

*(mostly improved font sizes) and believe that they are now clear and informative for the reader.*

Specific comments

Abstract

L15: It is unclear what "one year of open-water season" means. It would be helpful to add the dates and/or number of monitored days.

*Response: We agree that this was a bit unclear, in the revised version we have removed "open-water season" and also added "(in total 339 days excluding periods of ice and snow cover)" at the end of the sentence.*

L22: I recommend the authors to add a sentence about the effects of indeterminacy of streams draining agricultural catchments here, since this is an important finding of the paper.

*Response: We agree that this was missing. In the revised version we have added two sentences on this topic in the abstract.*

Introduction

L41-42: This sentence is unclear. What do you mean with positive and negative responses? Please clarify.

*Response: We mean that variations in stream CO2 concentration have been found to be both positively and negatively related to variations in stream discharge, i.e. either that CO2 concentrations increase when discharge does, or that CO2 concentrations decrease when discharge increase (dilution). This is now clarified.*

L44: "...dominant CO2 source areas of catchment soils"? Please rephrase this sentence.

*Response: This is now clarified.*

L45: Please specify what kind of other catchments.

*Response: There is no consensus (partly due to few existing studies) in which catchments CO2 are mainly controlled by hydrology or biology so it is hard to specify them more than catchments where the hydrological influence is low or non-existing. Hence, we keep the original formulation.*

L48: New paragraph needed.

*Response: Now added*

L50: Please specify what "relevant" time-scales are.

*Response: "(<hourly resolution)" is now added.*

L69: Please specify what you mean with high-resolution. Also, as mentioned before, it is unclear what "one year of open-water season" actually means.

*Response: Both "(hourly)" and "(in total 339 days excluding periods of ice and snow cover)" are now added to this sentence.*

Methods

L76: Please rephrase this sentence. Is it unclear if you mean the annual mean temperature or the January and July temperatures. This is especially important since you do not mention the precipitation in January or July - perhaps this could be added.

*Response: This is now clarified.*

L82: Stream pH ranging between 7.4 and 8.4. Also, this sentence would be much more readable if you would add a comma. In general, I would recommend using commas more frequently.

*Response: This is now clarified.*

L83-84: How much lower? Please provide a reference percentage.

*Response: It is hard to give exact percentages as the variables included in "nutrients" are many. We have instead added, as a general approximation, that the studied stream is within the $25^{th}$ percentile of the monitored agricultural streams in Sweden when it comes to DOC and nutrient levels.*

L86-87: This sentence could be moved to the beginning of the paragraph.

*Response: We agree and in the revised version have moved it to the start of the paragraph in ln??.*

L90: influences

*Response: Correct, now changed*

L91: Table S1; Figure S2

*Response: Correct, now changed*

L93: Would it be possible to add here the percentage that were snow/ice-free (and included in your study) as well as the percentage when the stream was falling dry?

*Response: We have again added the total number of measurement days. Concerning the number of days of the dry periods, this is more of a result and is already given in the text.*

L100: This is quite confusing for the reader, especially since you have not mentioned before that the stream is falling dry during some periods of the year. In general, I would recommend you to highlight the stream intermittency better, including adding some sentences in the introduction about this.

*Response: Again, the dry periods are here seen as result rather than description the methods. Still we needed to explain that analysis of the CO2 data was only made when runoff was generated. It would therefore not be logical to introduce the stream intermittency already in the introduction as this was not included in the aims of the study. However, as it became evident during the study that stream intermittency occurred and also was highly influential for the CO2 dynamics of the studied stream, this is something that needs to be discussed in a more extended way. See response to the first general comment.*

L108-109: Please clarify. What is the temporal resolution of your data?

*Response: As for many sensor-based systems averaging high-frequency data reduces the noise of the measurements and makes them more reliable. The given averaging time needs to account for relevant time-scales for the processes you want to study but also consider practical limitations as power consumption, data storage etc. In this case we measured at a 1 min interval and stored average values based on these 1 min measurements every 30 min (in 2017) or 60 min (in 2018). This is now clarified.*

L112-114: Please rephrase. Also, how many replicates?

*Response: We have clarified that one sample was taken at each sampling occasion in ln ??.*

L117: Please clarify. When was the phosphoric acid added?

*Response: The phosphoric acid was pre-injected in the vial before the sample was injected. This is already stated and so no change has been made.*

L121: Did you run any standards?

*Response: Yes, certified standards were analysed. This is needed as the DIC-values are given in relation to the PDB standard. This is clarified.*

L129: Add reference to Figure S2 here.

*Response: Correct, that is now added.*

L140: It would probably be easier to follow if you move this paragraph to the beginning of the methods section.

*Response: It is not clear how that this would clarify the text and make the methods more logical to follow. We prefer to keep this section where it is.*

L145: Another example of a sentence where the overall readability could be greatly improved if more commas are added.

*Response: We agree, and a comma has been added.*

Results

L155: This sentence is confusing. Precipitation is usually in mm/year however the period is for a bit more than a year. I assume that the "total precipitation" represent the precipitation for the whole period. Thus, it would be easier to read if the sentence would first state the mean air temperature (XXX) and then the total precipitation (XXX).

*Response: We write that "The mean air temperature and total precipitation for the entire period (Sep 26, 2017-Dec 12, 2018)". We believe this is already clear and have not made any changes.*

L205: It would be good to also add the corresponding pCO2 here for reference.

*Response: We have chosen to present the CO2 data as a concentration in the unit of mg C/L as this normalizes for solubility and makes it directly comparable with for example DOC/TOC concentrations if total aquatic C export would be of interest. We give corresponding pCO2 values in ln ?? as an example for how they compare. But we don't think*

*it is reasonable to give pCO2 values to all given CO2 concentrations in the manuscript while no addition has been made.*

L212: Same as above, the corresponding pCO2 values would be helpful as reference values.

*Response: Same as above*

Discussion

L231: "highly dynamic pattern in streamwater CO2 concentration".

*Response: Yes, we have added "concentration" for clarity.*

L250: Please add references.

*Response: Two suitable references for this statement are added.*

L256: could

*Response: We agree, is now changed*

L258-260: Please rephrase.

*Response: We have removed one piece of this sentence that might have been unclear.*

L266-270: Great paragraph. Would it be possible to develop more on this?

*Response: We have extended this paragraph to further develop the discussion about similarities/dissimilarities in carbon dynamics observed for agricultural streams.*

L271-272: Please rephrase.

*Response: The sentence is now rephrased in order to clarify.*

L309-311: Another great paragraph. This could also be further developed and better highlighted.

*Response: We thank the reviewer for this positive comment. We have in the revised version developed this section further and also highlighted this finding in the abstract and the conclusions.*

Tables

Table 1: Throughout the manuscript, you write either "land-use" or "land use". In the table it is obviously a spelling mistake; however, please be consistent with the terminology throughout the whole manuscript.

*Response: Thank you for noting, we have now used a consistent spelling "land use" throughout the ms.*

Table 2: Would be good to add the name of the catchment and not only the abbreviation.

*Response: The name is now fully spelled out.*

Figures

Figures: I recommend the authors to redo all figures. They are not intuitively designed or appealing for the reader.

*Response: Although the comment is very un-specific concerning what to improve, we agree that some of the figures needed polishing, especially concerning font sizes etc. We have updated many of the figures in order to make them easier to read.*

Figure 7: Add regression line?

*Response: Here we have used Spearman's Rank (which assumes a monotonic, non-linear, relationship), not regression, so fitting a line would not be appropriate. The given statistics in the figure refer to the Spearman rank test.*

Figure 10: In the text it is written that d13C-DIC was NOT a function of Q?

*Response: Yes, we write "Although there was a tendency towards more negative $\delta^{13}$C-DIC values at higher discharge, no significant relationship was found (Figure 10)". Although not significant from a statistical point of view, we still think it is useful information provided by the figure. One can imagine that the relationship might have been significant if the number of observations would have been more.*

---

## Author Comment (AC2) · 29 Mar 2020

**Reviewer (R#2) comments and author responses to ms bg-2019-486**

**Reviewer comments are given in normal style and with author responses in *italic***

Inland waters and specifically headwaters emit significant amounts of CO2 to the atmosphere; however, studies focusing in agricultural streams and including continuously measured in-situ CO2 from are rather rare. In this MS, the authors continuously monitored CO2 with cost-effective Co2 sensors during one year and explored the spatio-temporal variations of CO2 throughout the year as a function of hydrology and metabolism.

General comments

The MS bg-2019-486 provides an interesting study about CO2 dynamics in one stream draining a catchment largely dominated by agriculture. An important finding is that stream intermittency can cause rapid pulses of CO2 even in catchment with no pronounced dry and wet seasons. I think this is an important matter to better understand carbon emissions from streams at the global scale, in the context of climate change (change in hydrology). In line with this result, it could be useful to add somewhere in the discussion the spatial representativeness at the global scale of the stream studied here. In addition, it could be nice to add discussion/comparison of this agricultural stream with other agricultural streams worldwide, because the hydrology should be very different. To increase there adability, I suggest to better define some terms used in this study, particularly, open-water season, and the different periods, and also define better the time-intervals of these seasons throughout the text. Indeed, to my opinion, those terms are specific to boreal systems, and sometimes it is difficult to follow for a reader who is novice with boreal landscapes. A second important finding is the strong biologic control (aquatic primary production) of the CO2 dynamics during baseflow that should decrease CO2 emissions during this period. Indeed, during base flow it is common to observed higher CO2 concentration in streams because deeper levels of groundwater are involved. Perhaps the authors could further developed this. Overall, I found the dataset very interesting; it is rare to have such continuous measurements for CO2 in streams. In addition, I found the paper well written. Perhaps the quality of some figures could be improved. Overall, I support publication of this manuscript and below are some more detailed comments.

*Response: We thank reviewer #2 for their overall positive evaluation of our manuscript and are happy that publication in Biogeosciences is recommended after a revision. We believe that we in the revised manuscript have better discussed the spatial representativeness of our findings and also elaborated more on the primary production part. We further improved the quality of the figures (also in line with comments from R#1).*

Specific comments

Abstract

L. 15-16: It would be nice for the reader adding the size of the catchment, the date of open-water season, and the time-step of CO2 measurements.

*Response: We agree and have in the revised version added catchment size and total number of days of measurements. We have further replaced "continuous" with "hourly" to clarify the temporal resolution of the measurements.*

Introduction

L.31-33: The authors can check this reference that suit with their study (Deirmendjian et al, 2019. Importance of the vegetation-groundwater-stream continuum to understand transformation of biogenic carbon in aquatic systems – a case study based on a pine maize comparison in a lowland sandy watershed), where the concentration of CO2 in agricultural and forested streams (and in groundwater) in a temperate catchment was compared. They found no differences between both streams because degassing in agricultural streams was prevented.

*Response: We agree that this reference is very suitable and have added two sentences using information from it.*

L. 43-45: Please clarify this sentence. You mean that different level of soils are exported in function of the change in hydrology?

*Response: Yes, we mean that dependent on hydrological conditions different source areas in the catchment soils are hydrologically connected and contribute differently to the stream CO2. The variability in source areas are both vertically and laterally distributed in the soils and are hence activated differently dependent on groundwater position and dominating pathways. This pattern is further dependent on the catchment characteristics and land use. We have in the revised version tried to clarify the lateral and vertical consideration of source areas.*

L-40-55: To my opinion, there is a lightly lack of spatial references in this paragraph. Indeed, I guess that agricultural streams in tropical or boreal areas are very different in terms of hydrology and carbon dynamics. Could you mention spatial references?

*Response: We agree that the spatial coverage among the given references might look limited. We base this section solely on studies that have used high-frequency CO2 sensor data. This is now clarified. Also, the two references originally given (Dinsmore et al. 2013 and Crawford et al. 2017) both include data from multiple sites including boreal, temperate, alpine and subtropical areas. They further represent a large variety of forest, wetland and mountainous coverage. Hence, we believe that we have a relatively good spatial coverage, but to further support the tropical side we have added the very suitable Johnson et al. 2007 paper and adjusted the text according to this.*

L. 69: High-resolution: what is the time-step of measurements?

*Response: We have in the revised version added "(hourly)" after "high-resolution". Although we used 30 min resolution in 2017 and 60 min resolution in 2018 (in order to save power) "hourly" is likely the best option here. Further details on the different temporal resolutions are given in the method section.*

Methods

L.78: What kind of cropland it is? This is important for the d13C-DIC

*Response: The land is mainly used for cereal production and pasture. This clarification is now added.*

L. 83: Lower end: how much lower?

*Response: It is hard to give exact percentages as the variables including in "nutrients" are so many. We have instead added, as a general approximation, that the studied stream is within the 25th percentile of the monitored agricultural streams in Sweden when it comes to DOC and nutrient levels.*

L. 85: Growing season: what is the time interval?

*Response: The length of the growing season is on average ca 210 days starting in mid-April and ending in early November. This information is now included.*

L. 97: what was the concentration of gas standards?

*Response: Four standards were used (400, 1000 and 5000 ppm as well as 2%)*

L.100: discharge rates lower than 0 L/s: so you mean when the stream was dry or when the stream was frozen? Or both? It is a bit confusing.

*Response: This mean that CO2 data was just analyzed if runoff was generated over the V-notch dam i.e. excluding standing water or completely dry conditions. The instrument was never measuring during ice or snow conditions. This is now clarified.*

L.101: Figure S1

*Response: Yes, this figure reference is now given in ln??*

L109: You wrote one measurements each minute but then a temporal resolution of 30. It is a bit confusing what is the meaning of temporal resolution here?

*Response: As for many sensor-based systems averaging high-frequency data reduce the noise of the measurements and make them more reliable. The given averaging time needs to account for relevant time-scales for the processes you want to study but also consider practical limitations as power consumption, data storage etc. In this case we measured at a 1 min interval and stored average values based on these 1 min measurements every 30 min (in 2017) or 60 min (in 2018). This is now clarified.*

L.120: What is the volume of the injections?

*Response: The volume of the injections was 100 μL i.e. 7 × 100 μL per sample. This info is now added.*

L.129: Please specify that these streams were not located in your catchment and add the reference to the figure S2

*Response: This is now clarified.*

L. 145: Please define better your four periods. What are the time intervals?

*Response: We have now added number of days per period. We have also added a new table to the supplementary information that gives the full period description.*

Results

L.157: Please refer to figure 3

*Response: Figure 3 is now referred to.*

L.168-172: Please add corresponding pCO2 for reference, as you did

*Response: We have chosen to present the CO2 data as a concentration in the unit of mg C/L as this normalize for solubility and makes it directly comparable with for example DOC/TOC concentrations if total aquatic C export would be of interest. We give corresponding pCO2 values as an example for how they compare. But we don't think it is reasonable to give pCO2 values to all given CO2 concentrations in the manuscript while no addition has been made.*

L.166. To my opinion, I suggest to do that for the remainder of the text because pCO2 in ppmv is more "understandable" that CO2 in mg/L.

*Response: See comment above. We think this is also very much a matter of personal taste and as stated above we see clear advantages of presenting the CO2 data as concentrations rather than a volume fraction i.e. ppmv.*

 Discussion:

L.225: I would not rush on conclusion about zero/limited tree cover along agricultural streams, at the global scale. I am agree considering your figure S2 that this is the case in your catchment. However, in temperate climate it is very common to observe riparian forest along agricultural streams.

*Response: We agree that the statement was maybe too strong and have revised it. We still believe that canopy cover is important and that agricultural streams to a larger extent than for example forest streams are exposed to direct sun-light, even at the global scale.*

Figures

Figure 1: In the left part, I suggest to add a map of Europe rather than just Sweden. Please add a scale in the left part too.

*Response: We have updated the figure for the revised version.*

Figure 2: I suggest to separate the different periods (autumn, snowmelt, spring, dry period) with dotted lines, as you did in the next figure.

*Response: Good idea, this is now added*

Figure 4: It is not very intuitive what the time interval is for A, B, C and D.

*Response: The figure is now updated with appropriate font sizes*

Figure 5: Same remark

*Response: The figure is now updated with appropriate font sizes*

Figure 7: Perhaps add regression line with slope

*Response: Here we have used Spearman's Rank (which assumes a monotonic, not linear, relationship), not regression, so fitting a line would not be appropriate.*